biomechanics/behaviour

walking, terrain, ants, kinematics, roughness, *Linepithema humile*

**Authors for correspondence:**
G. T. Clifton
e-mail: glenna.clifton@gmail.com
N. Gravish
e-mail: ngravish@eng.ucsd.edu

# Uneven substrates constrain walking speed in ants through modulation of stride frequency more than stride length

G. T. Clifton[1], D. Holway[2] and N. Gravish[1]

[1]Department of Mechanical and Aerospace Engineering, and [2]Division of Biological Science, Section of Ecology, Behavior and Evolution, University of California, San Diego, USA

(iD) GTC, 0000-0002-5806-7254

Natural terrain is rarely flat. Substrate irregularities challenge walking animals to maintain stability, yet we lack quantitative assessments of walking performance and limb kinematics on naturally uneven ground. We measured how continually uneven 3D-printed substrates influence walking performance of Argentine ants by measuring walking speeds of workers from laboratory colonies and by testing colony-wide substrate preference in field experiments. Tracking limb motion in over 8000 videos, we used statistical models that associate walking speed with limb kinematic parameters to compare movement over flat versus uneven ground of controlled dimensions. We found that uneven substrates reduced preferred and peak walking speeds by up to 42% and that ants actively avoided uneven terrain in the field. Observed speed reductions were modulated primarily by shifts in stride frequency instead of stride length (flat $R^2$: 0.91 versus 0.50), a pattern consistent across flat and uneven substrates. Mixed effect modelling revealed that walking speeds on uneven substrates were accurately predicted based on flat walking data for over 89% of strides. Those strides that were not well modelled primarily involved limb perturbations, including missteps, active foot repositioning and slipping. Together these findings relate kinematic mechanisms underlying walking performance on uneven terrain to ecologically relevant measures under field conditions.

# 1. Background

The complex, three-dimensional structures of natural environments disrupt walking at multiple scales. Small-scale surface variation

(much smaller than body size) reduces foot adhesion, friction and contact area for walking animals [1–4]. By contrast, large-scale environmental structures (those far exceeding body size) present navigational challenges [5,6]. Far less is known about how substrate variability at intermediate scales influences locomotion. Human, animal and robotic studies of locomotion on substrates with three-dimensional variation at this intermediate scale often focus on discrete changes in substrate height, such as crossing a step or gap [7–11]. However, natural substrates present a continuous series of surface variations in both height and span, which probably induce distinct behavioural and kinematic adjustments. To investigate how naturalistic substrates with intermediate scale variation influence walking performance, we study one of nature's most proficient walking animals, the worker ant [12].

Non-flat substrates are commonly referred to as 'rough', 'uneven' or 'irregular' in animal locomotion experiments, with little standardized consensus in this terminology. In engineering fields, substrate variation is often called surface texture and the term 'rough' refers to variation at the nano- or micro-scale, and 'uneven' (or 'wavy') refers to variation on more macroscopic scales [13]. However, when associating animal locomotion to substrate, three-dimensional structures must be measured relative to body or foot size. A substrate experienced as 'uneven' for an ant functions as 'rough' for an elephant [14]. Here we define 'uneven' substrates as those with intermediate scale vertical and horizontal surface variations, at the order of an animal's body length. Soil particle size ranges from 0.002 to 80 mm in diameter [15] with southern California ant species ranging from 1 to 25 mm in body length [16], corresponding to three-dimensional substrate variations of 0.00008–80 body lengths.

Ant workers are wingless and can walk relatively long distances throughout their adult life [12], thus making them a good subject for understanding legged locomotion over uneven terrain. Ants sense and react to properties of the substrate that they traverse, as demonstrated through speed changes on uneven terrain [17–20] and through optimized foraging paths that respond to environmental changes [19]. Black garden ants show no preference for foraging on coarse versus fine sand [18], but seed harvester ants are more likely to drop or transfer seeds when walking on gravel versus sand [21]. A comparison of straight walking in nine ant species across flat, sand-covered and gravel-covered ground revealed net speed changes correlated with species and not leg length [20]. Although these studies suggest that substrate unevenness impacts ant foraging performance in some species, the findings are limited by (i) reporting only average speeds, (ii) not reporting limb kinematics, and (iii) using sandpaper or sand substrates, where increasing particle (grit) size causes an increase in both spanwise (horizontal) and vertical variation. We aim to isolate the influence of spanwise unevenness and directly relate it to detailed walking performance in ants.

Like many insects, ants generally walk using an alternating tripod gait, coordinating the movements of each middle limb with the fore- and hindlimb on the opposing side [22]. Studies in cockroaches and ants on flat ground have shown that walking speed directly increases with both step frequency and step length [23–25]. To accommodate movement on slopes or when carrying loads, ants can alter limb kinematics by modulating foot placement [26,27], the frequency and timing of stepping [28] and the contact forces produced by each foot [29,30]. However, similar deviations from stereotypical walking patterns have not been associated with spanwise substrate structure in ants. For cockroaches, running over a continuous array of blocks of varying heights reduces speeds by 20% and increases variability in body angular orientations. However, cockroaches do not appear to adjust foot placement of posterior legs to coincide with known good footholds by anterior limbs [31]. To our knowledge, outside of this study on cockroaches, insect limb kinematics has not been analysed on terrain that continuously varies in height. No studies have directly tested the impact of substrates that vary in spanwise unevenness, which probably constrains foot placement and disrupts horizontal positioning of the limbs.

Environmental cues and internal motivational states impact insect behaviour [32] and represent important factors for studies on locomotion [33]. Most biomechanical studies account for behaviour by inducing a consistent escape response or restricting analysis to a subset of behaviours (i.e. walking in a straight line or at a steady speed). But, quantifying locomotion performance across the natural range of behaviours is critical, especially when deciphering how substrate variability influences walking. Advances in new markerless and automated tracking methods accelerate data acquisition and analysis [34], empowering studies of natural walking that incorporate both environmental and behavioural variation. Using these techniques, we aim to comprehensively quantify normal walking kinematics while embracing behavioural variability.

Here we perform laboratory and field experiments to quantify the impact of uneven terrain on the walking kinematics and preferences of unrestrained Argentine ant workers (*Linepithema humile*). In laboratory experiments, we recorded thousands of videos of ants walking on 3D-printed flat and

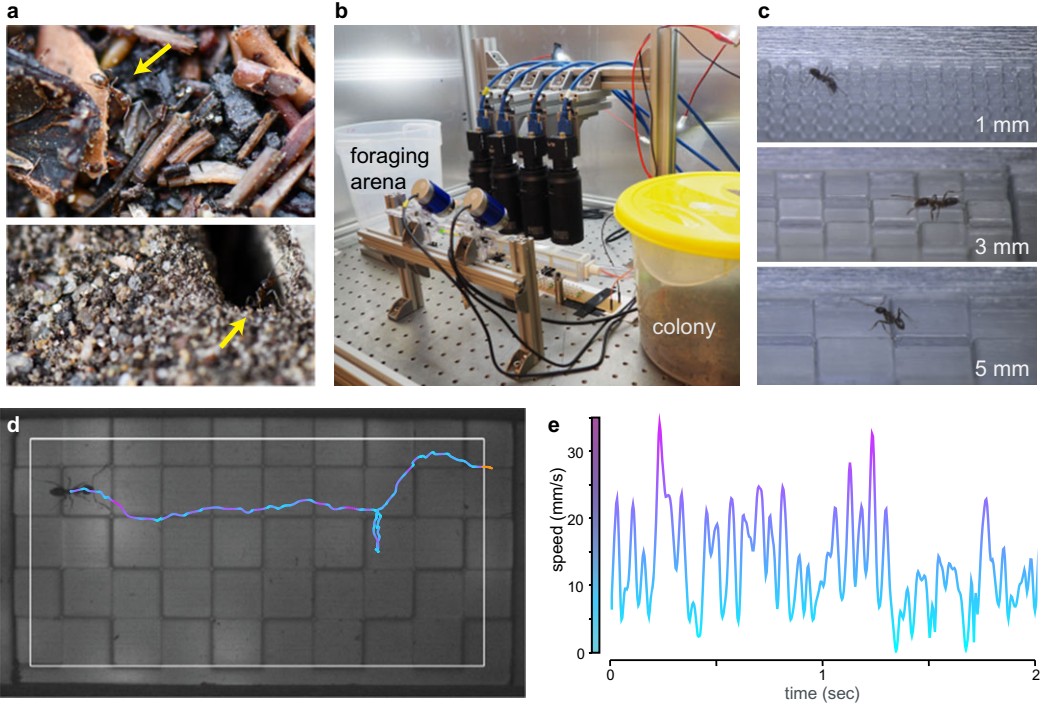

**Figure 1.** High-throughput camera system records colonies of Argentine ants walking over 3D-printed substrates that reflect natural unevenness scales. (*a*) Argentine ant natural habitats with terrain unevenness at varying scales, photographed at colony collection locations. (*b*) High-speed cameras recorded Argentine ant workers walking on flat and uneven substrates. (*c*) 3D-printed checkerboard patterns with a 1 mm step height and 1, 3 or 5 mm box width. (*d*) Any frames where the centre of the ant is within 60 pixels (approx. 1.9 mm) of the edge of the frame (white line) are removed from analysis (orange trace). (*e*) Ant walking speed varies within and between steps. Velocity trace does not represent full pathway displayed in (*d*). Colourmap represents walking speed in both panels.

uneven substrates with a horizontal scale approximately greater than, equal to, and less than worker body length (figure 1*a*–*c*). In outdoor experiments, we used the same substrates to test if colonies of ants would demonstrate a preference between uneven versus flat substrates. We hypothesize that if ants use cues such as optic flow to maintain foraging productivity then net walking speed should be conserved across different terrains. Alternatively, if uneven terrain challenges walking stability, we expect ants to decrease speeds in response to the frequency of encountering perturbations and to avoid terrains that induce greater instability. To further understand how walking performance shifts on uneven terrain, we used a deep-learning approach to track limb kinematics, identifying touchdown locations and timing. We used mixed effect modelling to identify how these gait parameters change with walking speed and compared how models trained on flat ground strides predict walking speeds on uneven substrates. If ants use a simple control strategy for walking, we expect their limb motions on uneven terrain to kinematically resemble walking on flat ground. However, shifts in relative timing and limb positioning could represent pre-planned or feedback-induced responses to walking instability. Our findings provide the first description of walking kinematics for ants on continuously uneven terrain while incorporating variability due to diverse, natural behaviours.

# 2. Results and discussion

## 2.1. Walking speeds in ants from laboratory colonies

We recorded 8266 videos of ants recruiting to food through a tunnel lined with randomly ordered substrates, including one flat and three uneven checkerboard patterns (electronic supplementary material, figure S1a–d). The checkerboard step patterns had a 1 mm step height and box lengths in three sizes: 1, 3 or 5 mm (figure 1*c*; electronic supplementary material, figure S1d). Tracking ant locations across frames (electronic supplementary material, figures S2 and S3), we find that ants walked more

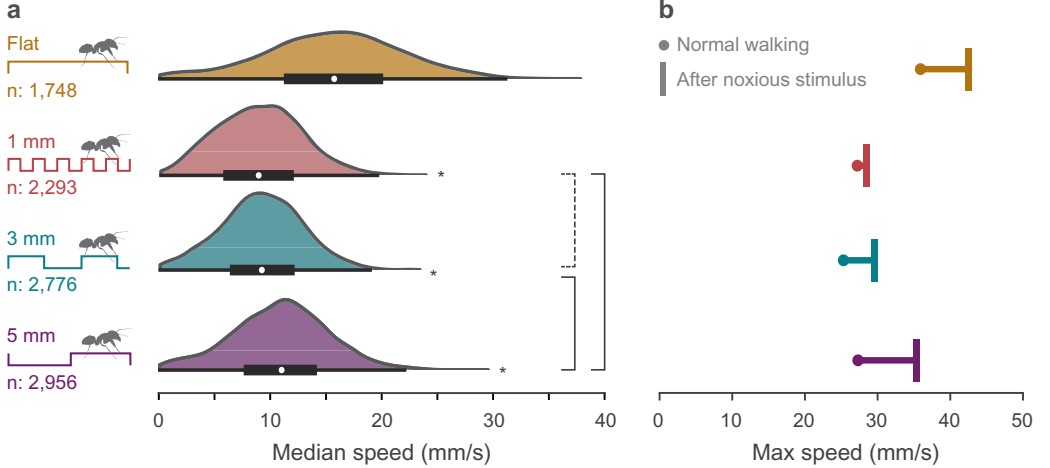

**Figure 2.** 3D-printed uneven substrates constrain average and peak walking speeds in ants. (*a*) Preferred walking speeds decreased on on uneven substrates. Speed distributions on all uneven substrates significantly differed from flat ground (asterisks, *p* < 0.0001, see S1 for statistical analysis) and from each other (solid brackets, *p* < 0.0001) except for 1 and 3 mm substrates (dashed bracket, not significant). (*b*) Maximum walking speeds were compared for unperturbed trials (dot) and those following a noxious airburst (T). Peak speed increased the least on the 1 mm substrate and most on the flat and 5 mm substrates.

slowly on uneven substrates. Instantaneous speed varied within and among steps (figure 1*d–e*; electronic supplementary material, figure S4), but median speed across the flat substrate was on average faster (15.1 mm s$^{-1}$) than on uneven substrates, particularly for the 1 and 3 mm checkerboards (8.7 and 8.9 mm s$^{-1}$, respectively) (figure 2*a*; electronic supplementary material, figure S5b).

Uneven substrates reduce walking speeds in ants with our results clarifying the impact of unevenness scale. Ants are known to slow down on uneven substrates that qualitatively differ (e.g. gravel versus sand, medium grass coverage versus bare ground) [18,21,35–37], but the only previous study that directly associated the scale of substrate structure to walking speed found that ants slow down when walking over particles greater than ⅓ body length [17]. All three uneven substrates tested in this study exceed this cut-off (worker body length approx. 3 mm), confirming that unevennesses at scales above ⅓ body length influence speed. However, our results show that preferred speed does not decrease linearly with unevenness scale and that walking speed is slowest at the intermediate size unevenness on our substrates. With similar average speeds on 1 and 3 mm substrates, there is not a distinct 'worst case' unevenness scale. It is unclear whether walking speed is a smooth function of ground unevenness, or if different unevenness scales induce distinct walking behaviours.

Because ants do not walk in straight lines or at steady speeds (electronic supplementary material, figure S4), focusing on the median speed across a given trackway masks important details of walking performance. Differences in average walking speeds between substrates could derive from variations in pathway sinuosity, behavioural modulation or physical speed limitations. To separate these factors, we first isolated sections of relatively straight walking (see Material and methods). In straight walking trials, we found a substrate-dependent pattern in average speed (electronic supplementary material, figure S5c) that aligns with the pattern found when including turning (electronic supplementary material, figure S5b), showing that turning is not responsible for speed reductions on uneven substrates. Second, we compared the distribution of speeds observed on each substrate to determine whether reductions in average speed resulted from a general shift in speed preference versus a transition to burst-and-stop walking. We found a unimodal speed distribution on each substrate (electronic supplementary material, figure S5d) suggesting that uneven substrates do not cause more frequent accelerations and decelerations, but instead a shift to overall slower speeds. Third, we tested the biomechanical limitations of walking on uneven substrates by inducing an escape behaviour [38]. A noxious jet of cinnamon-infused air injected into the tunnel every 5 min (electronic supplementary material, figure S1b) increased peak walking speeds on flat substrates by 6.9 mm s$^{-1}$ (19.7%) to a speed of 42.0 mm s$^{-1}$ (approx. 14 body lengths s$^{-1}$) (figure 1*e*; electronic supplementary material, figure S5b). However, 1 and 3 mm substrates constrained peak speeds to below 30 mm s$^{-1}$ (electronic supplementary material, figure S5b), with increases of only 1.7 and 4.6 mm s$^{-1}$, respectively. Compared to 5 mm substrates with an increase of 8.2 mm s$^{-1}$ in peak speed, substrates with an intermediate scale unevenness physically limited peak

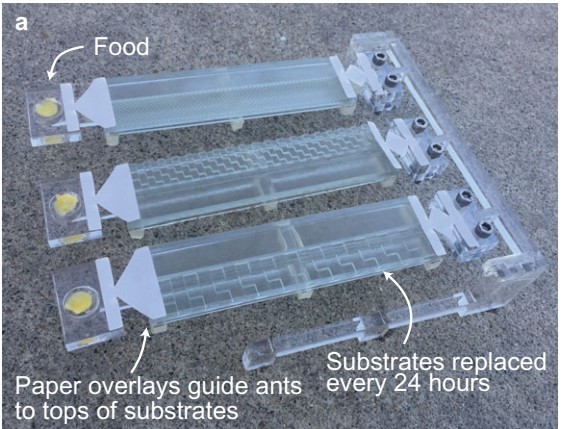

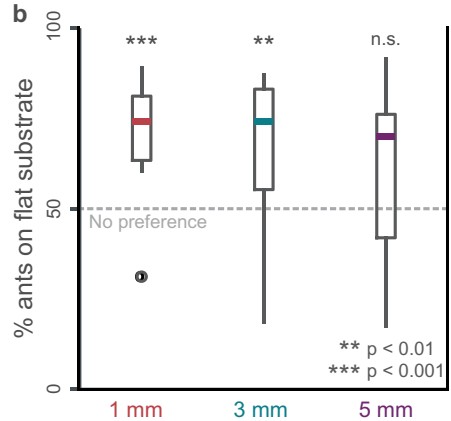

**Figure 3.** Ants prefer flat ground to 1 and 3 mm, but not 5 mm, checkerboard terrain (figure 1). (*a*) To test if ants perceive differences in walking performance due to substrate unevenness, an outdoor preference experiment tested whether ants would recruit to food sources over flat versus uneven substrates. (*b*) The percentage of ants on flat substrates for each 24 hour trial compared to 50% (dashed line), which represents no aversion to uneven substrates. Box plots correspond to median (centre line), upper and lower quartiles (box edges) and data range (whiskers). Significance values for each substrate and for comparisons among substrates are listed in electronic supplementary material, figure S1f.

walking speeds. Together these findings demonstrate that the observed reduction in average walking speed on uneven substrates does not result from turning, but instead results from a shift in preferred speed and a constraint on peak speed. Since ant walking speed directly relates to food acquisition rate [36] and the risks associated with foraging duration, such as predation [39] or desiccation [40], increasing substrate unevenness probably hinders colony-level performance.

## 2.2. Substrate preference in free-living ants

To complement our finding that uneven substrates constrain walking speeds in laboratory-housed ants, we tested whether ants sense and adjust navigation to avoid less favourable, uneven substrates in the field. We designed an outdoor preference experiment that allowed free-living Argentine ant colonies to recruit to food sources accessed via flat versus uneven substrates (figure 3*a*; electronic supplementary material, figure S1e; Movie S1). Our experimental design accounted for the influence of pheromone trails, navigational memory and edge-following behaviour [41]. Ants significantly preferred flat substrates over the 1 and 3 mm checkerboard patterns, but showed no preference on the 5 mm substrate compared to the flat substrate ($p < 0.001$, $p < 0.01$, and $p > 0.05$, two-sample *t*-test) (figure 3*b*; electronic supplementary material, figure S1f). Further, the magnitude of ant avoidance of uneven substrates aligned with the observed reductions in walking speed on those substrates (figure 1*d*).

The ability of Argentine ants to perceive relative substrate quality in the field and to adjust navigation accordingly probably contributes to colony-level performance. Previous observations that fire ants converge on trailways that minimize the time needed to walk across a boundary of two uneven substrates supports the notion that some ants use speed for path planning [19]. Similarly, path choice experiments in which an ant colony is given the option between two paths of differing length [5,42,43] show a preference for paths that minimize travel time. Here we confirm that when presented with multiple uneven substrates Argentine ants select paths that enable faster walking speeds. This finding directly links worker-level walking performance to colony-level foraging behaviour and supports the hypothesis that uneven substrates impact colony-level performance. Surprisingly, the selection of 'easier' paths is observed even over relatively short walking distances (12 cm) compared to Argentine ant foraging distances in the field (on average greater than 12 m and up to 63 m) [44]. More generally, this sensitivity to substrate structure may provide a critical link between the biomechanical challenges of locomotion on natural substrates and habitat preference in different species.

## 2.3. Ant limb kinematics

Up to this point, we have focused on average measures of walking performance on uneven substrates: speed and preference. However, we have not related walking performance to limb kinematics and

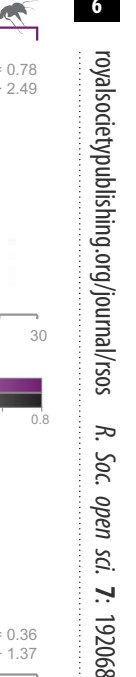

**Figure 4.** Speed, stride frequency and stride length relationships for walking on flat and uneven substrates. (*a*) The average instantaneous speed across a stride varies roughly linearly with stride frequency. Stereotypical flat walking strides were identified using a three-dimensional-density cut-off (grey solid and dashed polygons). (*b*) Stride length remains relatively constant across average stride speeds. For both (*a*) and (*b*), data plotted in colour represents strides well-modelled by the 'full' linear mixed effects model generated from stereotypical flat walking strides ('well-modelled strides', see Material and methods for details). Grey data represents strides that were poorly modelled (outlier strides).

disruptions from steady walking patterns. Like many insects, ants generally walk using an alternating tripod gait, coordinating the movements of each middle limb with the fore- and hindlimb on the opposing side [22]. To accommodate varying external conditions, such as climbing vertically or carrying a load, ants alter limb kinematics by modulating foot placement [26,27], the frequency and timing of stepping [28] and the contact forces produced by each foot [29]. These shifts in limb motion, coordination and dynamics offer an opportunity to identify disruptions from normal walking and to test whether these disruptions dictate overall performance on uneven substrates.

Traditionally, biomechanical studies of limb motion required manual tracking, with the time intensity of this approach precluding analysis of the large datasets needed to capture the variability of freely behaving animals. Advances in new markerless and automated tracking methods accelerate data acquisition and analysis, enabling our assessment of walking kinematics in freely recruiting ant workers. To track limb movements in our full dataset of videos, we used a deep-learning approach. But because walking on uneven substrates includes considerable kinematic variability, we performed extensive post-processing to ensure high confidence in the automated tracking (electronic supplementary material, figure S7). We analysed tarsal motion with respect to the substrate to identify touchdown (TD) timing and location (electronic supplementary material, figure S8), with sequential TDs defining strides for each limb. Comparing our computational identification of TDs to a subset of manually tracked videos on each substrate, our approach correctly identified 239/244 TDs (98%), with an average timing offset of less than 1 frame for all substrates. With this approach, we analysed more than 11 700 walking bouts, corresponding to approximately 2 500 000 video frames.

In legged locomotion if slipping does not occur, an inter-relationship exists between stride length, stride frequency and speed. Scaling relations derived and confirmed in vertebrates and several invertebrates [23–25,45] predict linear relationships in both speed versus stride frequency and speed versus stride length. We confirm a largely linear stride frequency relationship in Argentine ants walking on flat ground ($R^2 = 0.91$, figure 4*a*), but without an observed plateau at a maximally sustained stride frequency. According to allometric scaling equations [45], this transition should occur in Argentine ants

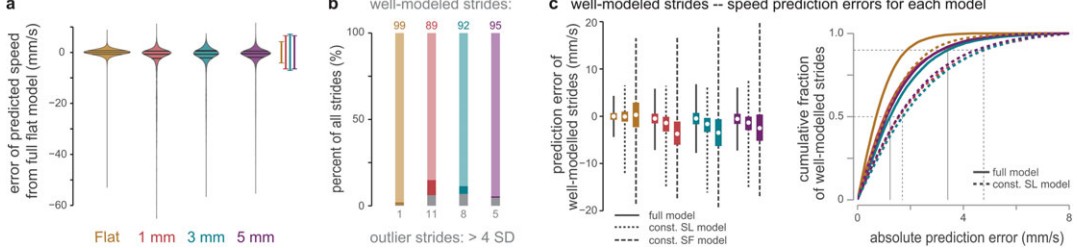

**Figure 5.** Modelling of walking speed using stride frequency and stride length from stereotypical strides on flat ground. (*a*) A full model using both stride frequency and length predicted most strides with a small error from observed values. The highest density regions of these distributions were used to determine the cut-off for well-modelled (brackets) and outlier strides on each substrate. (*b*) The percentage of strides that were well-modelled (bars in colour) versus outlier (grey bars) for each substrate. The darkest, colourful portion of each represents the range in percentages across all seven lab colonies. (*c*) The error between predicted and observed well-modelled speeds for the full model (solid lines), constant stride length model (dotted lines) and constant stride frequency model (dashed lines) on each substrate. Box edges represent quartiles with whiskers representing the full range. A cumulative density plot shows the fraction of strides versus prediction error. Thin, solid (full model) and dashed (constant stride length model) lines represent the prediction error cut-offs for 50% and 90% of strides.

(mass = 0.43 g) at a stride frequency of 25 Hz and speed of 78.6 mm s$^{-1}$, well above the observed data range (40 mm s$^{-1}$).

The stride lengths of ants walking on flat ground primarily range from 1.8 to 2.6 mm, but large relative variations obscure a tight correlation with speed (figure 4*b*). The slope of the relationship between stride length versus speed is 0.03 s, which falls within the range of published values for ants [25,46], cockroaches [47] and fruit flies [48–50]. Given our large sample size, we rely more on effect size and the coefficient of determination ($R^2$) instead of *p*-values to represent the regression [51,52]. The relationship between walking speed and stride length exhibits considerable variation ($R^2 = 0.50$). Historically, a large body of research reports a tight linear regression between stride length and speed in insects [22,24,25]; however, recent studies that incorporate behavioural variability and large sample sizes corroborate a weak correlation [48,53,54]. The weak correlation we find between speed and stride length is not due to turning since we restricted our analysis to strides with a heading of less than 15°, but could derive from unsteady walking (e.g. accelerations) present in unconstrained experimental conditions. These results suggest that stride frequency predicts walking speed more closely than stride length, implying that ants may primarily modulate step timing to control speed.

To more directly test how stride length versus stride frequency contributes to ant speed while accounting for potential confounding parameters, we compared three linear mixed effects models applied to 'stereotypical' strides from ants walking on flat ground (figure 4*a*; electronic supplementary material S9). The full model included both stride length (SL) and stride frequency (SF) as factors, with the other two reduced models each lacking one of those variables ('constant SL' and 'constant SF'). Comparing model fit between the full and reduced models showed that the constant SL model is 3.5 times more similar to the full model compared to the similarity between the constant SF and full models (chi-squared values of 14 644 versus 53 301 for constant SL and constant SF respectively) (electronic supplementary material, table S1). While the exclusion of either factor caused a significant difference from the full model ($p < 0.001$), our large sample size contributes to this finding.

As a secondary assessment of the full and reduced models, we used each model to predict the stride speed for all strides on all substrates then compared these values to the observed speeds. In addition to evaluating relative model performance, this approach enables comparisons of walking strategies across substrates. The error distributions between the predicted and observed speeds for the full model consisted of two parts, which we isolated to analyse separately (figure 5*a*). A largely normal portion of the distribution defined the 'well-modelled strides', while a long tail of 'outlier strides' contained points that largely deviate from the observed speeds. The percentage of strides in each category varied by substrate, with well-modelled strides representing between 89 and 99% of all strides (figure 5*b*).

Focusing on the well-modelled strides, the full model predicted stride speed on average within 0.56 mm s$^{-1}$ on all substrates (figure 5*c*). This level of congruity is expected for strides on flat ground, since the model was generated from a large portion of those strides. But surprisingly, the full flat ground walking model also predicted the forward speed of an ant walking on uneven substrates within a small

margin. Over 50% of the well-modelled strides on all substrates were predicted within 1.5 mm s$^{-1}$, with 90% predicted within 3.5 mm s$^{-1}$ (figure 5c). Accounting for the absolute observed speeds for each stride (with most strides between 5 and 20 mm s$^{-1}$ on uneven substrates), more than half of all well-modelled strides had a predicted velocity within 10% of the observed velocity. In comparison, the constant SL model predicted stride speed on average within 1.6 mm s$^{-1}$ and the constant SF model predictions averaged within 3.9 mm s$^{-1}$. All models performed best for strides on a flat substrate and worst on 1 mm substrates. The constant SF model did not predict stride speed as well as the constant SL model, as demonstrated by a larger median and broader range in prediction error (figure 5c). The constant SL model predicted over 50% of all well-modelled strides within 1.8 mm s$^{-1}$ and 90% of strides within 4.8 mm s$^{-1}$ (figure 5c), compared to 4.3 and 9.0 mm s$^{-1}$, respectively for the constant SF model (electronic supplementary material, figure S10).

The accuracy of simple models in predicting stride speed based on stride frequency and/or stride length reveals two main findings of this study. First, we found that stride frequency predominantly dictates stride speed in ants, with stride length remaining relatively constant. Second, we found that most strides on uneven substrates resemble walking on flat ground. Previous observations in ants and other insects show that stride length increases linearly with speed, though at a slower rate than stride frequency [22,24,28]. Here we confirmed and extended this pattern, showing that stride frequency alone is sufficient in predicting stride speed despite a low correlation between stride frequency and length (electronic supplementary material, table S1). Therefore, ants walking on uneven substrates shifted to slower stride frequencies, accounting for the observed slower speeds.

Surprisingly, the dependence of speed on stride frequency over stride length was retained for walking on uneven ground in ants. When confronted with uneven terrain, humans generally reduce step lengths [55,56], particularly in older adults or in individuals with compromised gaits who must compensate for reduced balance [57,58]. Although ants slowed down on uneven substrates, these speed reductions were primarily modulated by changes in stride frequency. It is unclear exactly why ants decrease stride frequency on uneven ground though this could derive from both biomechanical or behavioural factors. Biomechanically, ants that attempt to move limbs at higher frequencies could experience detrimental foot–ground interactions that disrupt walking, such as foot slipping or missteps, which result in lower stride frequencies. Furthermore, the limbs of small animals, like ants, have less inertia and actively control stride frequency throughout the stride compared to larger animals that rely on momentum [59]. Behaviourally, ants may decrease stride frequency on uneven terrain to provide more time for sensory feedback or to slow limb motions in an attempt to proactively avert disruptive foot–ground interactions. Such proactive modulations of frequency may be a plastic response based on previous experience, which could be observed in future studies of learning trials or transitions from flat to uneven ground.

## 2.4. Characterization of highly divergent strides

The full model generated from stereotypical strides on flat ground accurately predicted the average speed for most strides on all substrates, but a portion of strides strongly deviated from the model predictions. These outlier strides only represented 1% of the strides on flat ground, but ranged up to 11% on 1 mm substrates (figure 5b). Accounting for six limbs, an 11% outlier rate corresponds to an average of at least one outlier step every two strides. These strides generally have a slow average speed and short duration (high frequency) (figure 4, grey), but what walking behaviours cause outlier strides? We compared foot touchdown position for well-modelled and outlier strides on each substrate, finding highly consistent foot placement on flat ground during well-modelled strides (figure 6a). Surprisingly, the preferred well-modelled locations were tightly conserved across flat and uneven substrates despite increased variability on uneven terrain (figure 6a,b). However, foot placement in outlier strides on all substrates shifted posteriorly and medially (figure 6b), coinciding with the swing trajectory of each foot. The average placement of the feet during outlier strides diverged from the well-modelled preferred positions by 0.34–0.45 mm (figure 6c). Using a logistic regression, we predicted the likelihood of outlier strides based on foot displacement (figure 6d). To be identified as an outlier, strides required a larger displacement on flat ground compared to uneven substrates. The displacement pattern among uneven substrates mirrored the patterns in both speed reduction and outlier stride frequency, with the most extreme condition occurring for the 1 mm checkerboards, while the 5 mm checkerboards approached flat ground values. These findings confirm that most strides on uneven substrates resemble walking on flat ground, with those strides that diverge from this walking pattern associated with touching the foot down away from the normally preferred positions.

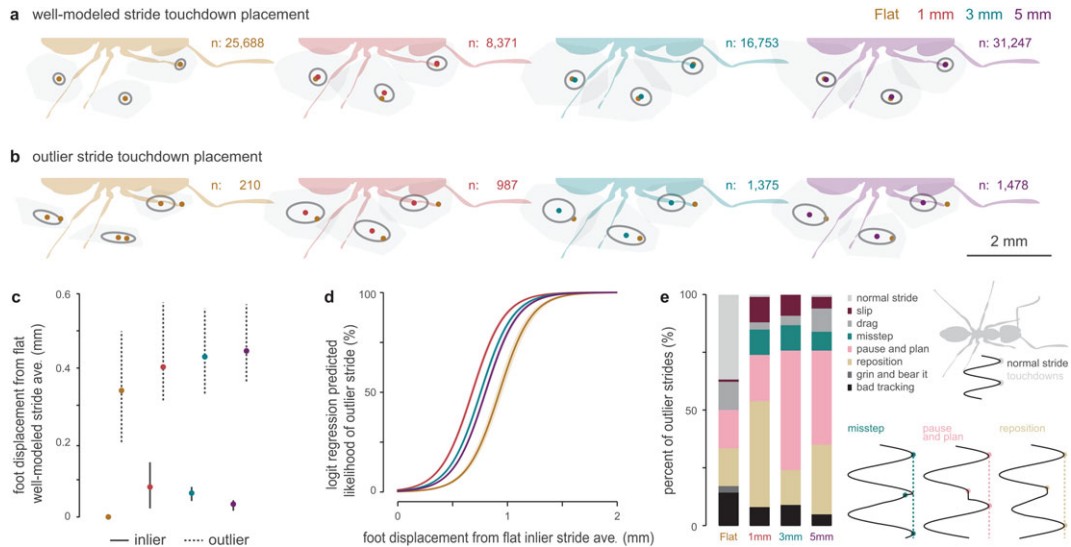

**Figure 6.** Touchdown foot placement and manual characterization of outlier strides. (*a*) Average placement of the foot at touchdown for well-modelled strides on each substrate. Circles represent one standard deviation of the two-dimensional distribution. Yellow data points represent the average location on flat ground. (*b*) Average placement of the foot at touchdown for outlier strides on each substrate. Circles represent one standard deviation of the two-dimensional distribution. Yellow data points represent the average location of well-modelled strides on flat ground. (*c*) Total displacement of average foot position relative to average location of well-modelled strides on flat ground. Whiskers represent the range across limbs, circles represent averages. (*d*) Likelihood of identification as an outlier stride versus displacement for each substrate. Curves were generated using a logistic regression. (*e*) Manual characterization of outlier strides on flat and uneven substrates. Diagrams represent the anterior–posterior trajectory of the tarsus with respect to the ant (*x*-axis) across time (*y*-axis). Dots represent touchdowns.

The 'divergent' strides identified through our modelling could result from multiple kinematic behaviours such as slipping [60,61], premature foot touchdown [62] and active repositioning of the limb during stance for a more favourable foothold [63,64]. To identify the behaviours underlying divergent strides, we manually characterized 100 randomly selected outlier strides for each substrate (electronic supplementary material, figure S11). This characterization represents a first qualitative assessment of outlier strides, motivating future quantitative analyses using gait phasing [10]. We find that on flat ground, with only 210 identified outlier strides, more than half corresponded to normal walking or automated tracking errors (figure 6*e*). Since the tracking error strides represent less than 0.06% of all flat strides, we do not believe that they invalidate our methods or general findings. The remaining flat outlier strides mostly consisted of dragging the foot forwards during stance, repositioning of the foot after a collision, or a brief premature touchdown of the foot as the ant slows down and appears to choose its next step (pause and plan). On uneven substrates, outlier strides also included missteps, when the foot touches down after reaching and missing its targeted foot placement. Notably, walking on the 3 and 5 mm substrates induced a high percentage of pause and plan strides. These strides mostly coincided with the ant reaching the edge of a checkerboard box and tapping the foot (often of the forelimb) along the corner before placing it forward. Insects possess proprioceptive sense organs within each limb, including both muscle strain sensors and campaniform sensilla in the cuticle, which have been shown to coordinate limb movements, control posture and detect foot attachment to the ground [65,66]. Most of the studies on insect proprioception focus on insects in controlled conditions or walking on flat ground. The pause and plan strides identified here represent a potential mechanism for ants to use proprioception while walking on uneven terrain.

## 3. Conclusion

Our findings demonstrate that ants are able to mostly conserve flat ground walking patterns and maintain preferred foot placement while on naturally uneven substrates. Even though ants reduced walking speeds substantially on uneven substrates, over 89% of their strides were accurately modelled based on flat walking data with speed primarily governed by stride frequency over stride length. To diminish the influence of the

substrate irregularities, ants may use specialized adhesive tarsal pads that secure foot attachment [29] and benefit from polypedalism, which improves passive recovery from impulsive disruptions [9,10,67]. With neuromechanical and robotic control research often focusing on accurate foot placement [68] or navigational planning [69], our findings highlight the potential for developing decentralized mechanisms for navigating uneven terrain. Further, we show that ants in the field seek out substrates that enable faster walking speeds. By extending the large-scale, automated approaches established in this study to multiple species and field conditions, it should be possible to uncover how walking strategies in ants contribute to foraging patterns, habitat selection, assemblage structure and evolutionary diversification.

# 4. Material and methods

## 4.1. Experimental design: general set-up, ant collection and recording

To determine how the Argentine ant walks on uneven ground, we recorded workers from laboratory colonies walking over 3D-printed substrates. Between March and May of 2018, material for laboratory colonies was collected from nine locations on the campus of the University of California, San Diego, with each location separated by at least 500 m (electronic supplementary material, figure S1a). After finding a large aggregation of ants (more than 1000 individuals, including some workers carrying brood), we excavated approximately 300–600 workers, along with brood and queens, and placed them in a lidded plastic container. The data from one colony was removed from analysis since many ants escaped during the recording session.

Laboratory ant colonies were housed in a container within a custom recording enclosure on a 12 : 12 hour light : dark cycle (for details, see electronic supplementary material, S1, Appendix A). To reach food, ants walked through a 3D-printed tunnel (Connex3 Objet 350, VeroClear material, Stratasys Inc., USA) (figure 1b) with floor openings for four 3D-printed substrates randomly positioned within the tunnel (figure 1c). The substrates measured $16 \times 30$ mm and consisted of a flat surface and three checkerboard patterns of a 1 mm step height and a square edge of 1, 3 or 5 mm (electronic supplementary material, S1, Appendix A).

The ceiling and side of the tunnel incorporated microscope coverslips ($24 \times 50 \times 0.25$ mm, AmScope, Ltd, USA) to enable filming. Four machine vision cameras (Blackfly S 13Y3M, PointGrey, Inc., Canada) and lenses (20–100 mm, 13VM20100AS, Tamron Co., Ltd, Japan) recorded from above the tunnel. Two webcams (YoLuke A860-Blue, Jide Technology Co., Ltd, China) filmed from an oblique overhead angle to identify when ants approached a substrate. The tunnel was backlit using a custom array of infrared LEDs (940 nm). During two 4 hour recording sessions for each colony, the machine vision cameras recorded ants for 3 s at 240 fps (720 frames total, $1000 \times 550$ pixels). After recording, each machine vision camera paused for 40 s to reduce the probability of re-recording the same individual. In total, 8266 videos were recorded across eight colonies.

## 4.2. Experimental design: noxious stimulus procedure

To elicit maximal walking performance, one colony was exposed to a repetitive noxious stimulus. Based on a study that found an aversion to the scent of cinnamon in ants [70], we built a custom system to inject cinnamon-infused air into the tunnel. Pressurized air (15 PSI) was infused with cassia cinnamon oil (Healing Solutions, LLC., USA). The opening of an arduino-controlled solenoid valve allowed the cinnamon air to flow through a manifold to four tubes inserted into the wall of the tunnel above each substrate. Each noxious stimulus consisted of two 0.25 s bursts separated by 1.00 s. After unperturbed walking was recorded for an hour during each session, cinnamon bursts were triggered every 5 min throughout the remainder of the recording session.

## 4.3. Tracking: automated full-body tracking

Tracking ant whole-body motion was achieved by identifying all ants in each frame and associating individuals across frames. To identify ants in each frame, we (i) normalized the video using background division, (ii) used image processing to isolate the body of each ant, and (iii) fit contours to each ant body to estimate the location and orientation (electronic supplementary material, figure S2). For details on each step, see electronic supplementary material, S1, Appendix B. Ants identified in each frame were associated across frames using a Kalman filter [71].

## 4.4. Analysis: average ant speed post-processing and statistical analysis

In Python, the x- and y-coordinates of the kalman-associated ant 'trackways' were filtered using a low-pass butterworth filter (Scipy, $n = 2$, $\omega_n = 0.2$) and differentiated to calculate the instantaneous velocity. Any data points where the centre of the ant was within 60 pixels of the edge of the frame were removed. To estimate the average horizontal walking speed during a trackway required accounting for behavioural variation, such as antennal cleaning or ant interactions. To identify and remove these stationary behaviours, we calculated the net distance travelled over 90 frames (0.375 s) centred around each time point and removed any datapoint when the ant had moved less than 50 pixels (1.56 mm) during those 90 frames (electronic supplementary material, figure S4). This cut-off was manually validated in 20 trials.

Median walking speeds were calculated using these processed speed profiles (electronic supplementary material, figure S4). The distribution of median speeds was compared across substrates using linear mixed effects modelling in R (figure 1d; details in electronic supplementary material, S1, Appendix C). Briefly, a chi-squared likelihood ratio test determined whether substrate type significantly impacted walking speed and then estimated marginal means of the model were used to calculate the significance of comparisons among substrate types.

Slower walking speeds could result from turning more frequently or more acutely on one substrate. To account for this potential compounding factor, speed was also analysed for straight walking strides only (with a heading <15°, see below on 'Tracking: touchdowns and strides'). The net distance travelled during each straight stride was divided by the stride duration. We applied the above linear mixed effect model approach to the straight stride dataset (electronic supplementary material, figure S5).

## 4.5. Analysis: noxious stimulus

Comparing median speeds for each trackway did not fully encapsulate walking performance. To further clarify how uneven substrates impact walking, we calculated the distribution of walking speeds used by ants on each substrate. We applied this analysis to the eighth collected colony, including ants walking before and after a noxious burst of cinnamon-infused air. We accounted for within-step accelerations by calculating a windowed average speed (electronic supplementary material, figure S4). Instead of using a time-based histogram of the instantaneous speeds on each substrate, which over-represents slow speeds, we compared the distance travelled at each speed. We restricted our analysis to trials with at least 50 frames (0.21 s) of processed data.

The distribution of distances travelled versus speed was compared for trackways during the first hour of recording on each day (no noxious stimulus), for trackways recorded less than 2 min after a cinnamon airburst and for trackways recorded between 2 and 5 min after a cinnamon burst (bursts were every 5 min) (electronic supplementary material, figure S6). For flat/1 mm/3 mm/5 mm substrates, the total travelled distances compared were 295/485/378/128 cm. The median speed was identified by finding the speed cut-off with 50% of the total distance travelled at faster speeds. Peak speeds were identified using speed cut-offs of 5% and 2%.

## 4.6. Experimental design: outdoor preference experiments

To test whether ants demonstrated an awareness of and preference for substrate unevenness, we conducted choice experiments in two locations near the UCSD campus in La Jolla, CA (electronic supplementary material, figure S1a). Long rectangular sections (120 × 15 mm) of each substrate type (flat, 1, 3 and 5 mm) were 3D-printed and attached to a custom cantilever-support structure that held three pairs of rectangles (electronic supplementary material, figure S1e, see S1 Appendix D). Each pair of rectangles compared a flat versus uneven substrate, therefore, each set-up tested 1, 3 and 5 mm substrates simultaneously. The end of each pair of substrates contained ant food based on the Bhatkar–Whitcomb diet, combining hard-boiled egg yolk, sugar and water [72]. To discourage ants from walking on the side of the cantilever-support structure or randomly favouring one substrate, a diamond-shaped piece of paper was attached on top of the plank leaded to each pair of substrates.

We recorded ant substrate preference using the cantilever set-ups for three weeks during October and November of 2018. A time-lapse camera (Re 16MP camera, HTC, Taiwan) positioned above each set-up recorded images every 3 min. After 24 h of recording, each set of substrates was replaced (in randomized order) before being soaked in a mixture of windex and water for the next 24 h. This mixture was used to remove all pheromone trails from the substrates before the next recording session. The time-lapse videos of the outdoor preference set-ups (electronic supplementary material, figure S1e) were automatically

analysed to compare the number of ants on flat versus uneven substrates. The total percentage of ant pixels on smooth versus uneven substrates was calculated and compiled for each day of recording (for details, see electronic supplementary material, S1, Appendix E).

Ants did not discover the outdoor set-up three times, probably due to the onset of cooler conditions in November. Of set-ups that were discovered, the food sources were discovered 93% of the time, with ants ignoring the substrate bridges once on 3 mm bridges and twice on 1 mm bridges. Across both set-up locations, ant preference was tested 13, 14 and 15 times for 1, 3 and 5 mm substrates, respectively. For each substrate comparison, we performed a two-tailed T-test to determine whether the percentage of ants choosing to walk on a flat versus uneven substrate differed from no-preference, 50% (scipy.stats.ttest_1samp) (electronic supplementary material, figure S1f). To test if ant preference differed between each combination of uneven substrates, we calculated a two-tailed Welch's T-test for two independent samples with unequal variances and sample sizes (scipy.stats.ttest_ind).

## 4.7. Tracking: 'LEAP' deep-learning tracking

Manually tracking the limbs in 8000+ videos is not feasible. Instead, we used a recent deep-learning approach, implemented in Matlab and Python [34]. The LEAP deep-learning workflow tracks a set of user-defined points on square videos centred on a subject. The user determines a skeleton of connected points to track then iteratively hand-labels frames and predicts tracked point locations on new frames to eventually build a robust training set of labelled frames for the final prediction model. Our training set of 679 manually tracked frames generated a model to predict 10 body landmarks, including the location of each tarsus, in novel frames. This approach worked extremely well for ants walking on flat substrates, but decreased in accuracy for the variability associated with uneven substrates. In order to implement this process with our highly variable data, we (i) produced trustworthy input ant-centred videos, (ii) hand-digitized an extensive training set of 679 frames, and (iii) implemented post-tracking confidence checks to improve or remove untrustworthy data. Details of these methods can be found in electronic supplementary material, S1, Appendix F.

By visually inspecting the raw LEAP and post-processed tracking for 30 recorded videos, we confirmed that our approaches were conservative in removing any potentially inaccurate tracking. In total, 11 700 trackways—corresponding to approximately 2 500 000 video frames—were passed through the LEAP tracking and custom post-processing workflow.

## 4.8. Tracking: touchdowns and strides

The deep-learning-based LEAP tracking resulted in time-varying locations of the thorax, neck and tarsi. These data were used to identify when each foot touched down onto the substrate and the strides between these TDs. Because the movement of the body during stance often occluded fore and mid-limb toe-off, we were unable to reliably determine stance and swing timing.

The timing and location of TDs were determined using tarsal speed, but kinematic variability and discontinuous tracking (particularly on uneven substrates) required additional measures to ensure accurate TD detection. For details see S1, Appendix G. To test the accuracy of this approach, we compared our computationally identified TDs to visually identified TDs in 18 videos, corresponding to over 57 TDs per substrate type for a total of 244 TDs. Of these TDs, 239 (98%) were correctly identified within 10 frames, with an average offset of one frame (4.2 ms). Of the five TDs identified computationally but not manually, three were confirmed having been missed during the initial manual tracking. The remaining two were false identifications by our post-processing approach.

Strides were defined as occurring between two trusted TDs with two conditions to ensure no intermediate TDs were missed (see electronic supplementary material, S1, Appendix G). We calculated four variables for each trusted stride. (i) *Stride frequency* was calculated from the inverse of stride duration (the number of frames between TDs). (ii) *Stride length* was defined as the distance between the location of the foot at the beginning and ending TDs. (iii) The *average stride speed* was the mean of the instantaneous velocity's forward component (aligned with the ant's anterior–posterior axis). (iv) The *travel direction* of the ant was the net angular displacement of the thorax of the ant during the stride relative to the ant's facing at the beginning TD. All strides with a travel direction <15° were classified as straight walking.

The LEAP tracking of body and limb movements quantify the horizontal, two-dimensional component of motion. However, the ants in our experiment walked over three-dimensional terrain, potentially reducing the accuracy of our measurements. Given an average stride length of 2 mm, a 1 mm vertical height difference between stride touchdowns would result in an underestimation of

stride length by 0.23 mm. This discrepancy could be responsible for some of the data spread observed in figure 4*b*; however, analysing horizontal stride length in relation to horizontal body speed reduces the impact of this discrepancy.

## 4.9. Analysis: average stride speed modelling

To test whether ants use stride frequency or stride length to modulate walking speed, we compared three linear mixed effects models (for details see electronic supplementary material, S1, Appendix H; figure S9). The source data were drawn from straight strides on flat ground, while removing unusual strides. The full model included both stride frequency and stride length as fixed effects to explain average stride speed. Two reduced models either held frequency or length constant. Each model was used to predict the average stride speed for all strides on all substrates, then we analysed the errors between predicted and measured speeds. The most dense points of the full model error distribution were used to define a normal distribution, separating well-modelled and outlier strides with a cut-off of four standard deviations (see electronic supplementary material, S1, Appendix H). The well-modelled stride errors were compared using boxplots and an empirical cumulative distribution function.

Foot placement at touchdown was compared for well-modelled and outlier strides on each substrate. To find the average placement and its variance, we used a principal components analysis (sklearn.decomposition.PCA). The standard deviation of the first two principal components defined the ellipses in figure 5*d,e*. For the first TD of every stride, we calculated the net displacement from the preferred foot placement of well-modelled strides on flat ground. A logistic regression (see electronic supplementary material, S1, Appendix H) was used to predict the probability that a stride was an outlier based on its displacement.

To identify the behaviours associated with outlier strides on each substrate, we manually observed and classified a random subset of these strides. We randomly selected 100 outlier strides from each substrate type and each of eight stride classifications. Two categories involved false identification (normal strides or bad tracking), two represented disruptions during the stance phase (slips and drags), and four represented disruptions during the swing phase (misstep, pause and plan, reposition, grin and bear it) (electronic supplementary material, figure S11).

Data accessibility. Relevant code for this work are stored in GitHub: https://github.com/gtclifton/AntTrackingCode and have been archived within the Zenodo repository: https://zenodo.org/badge/latestdoi/167629203. The datasets generated during and analysed during the current study are available within the Dryad Digital Repository: https://doi.org/10.6075/J0RR1WM2 [73].

Authors' contributions. All authors contributed to the conceptualization and methodology design for this study. G.T.C. performed data collection, analysis and drafted the manuscript. All authors contributed to critically revising the manuscript. All authors gave final approval for publication and agree to be held accountable for the work performed therein.

Competing interests. We declare we have no competing interests.

Funding. Funding support for this research was provided by the Army Research Office under grant W911NF-17-1-0145 and a UC San Diego Chancellor's Research Excellence Scholarship.

Acknowledgements. We thank Prof. Michael Tolley for 3D printer access and Ben Shih and Chris Cassidy for printing help. We thank Axel Qin for help in experiment prototyping.

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
