## [Reviewer comments · Royal Society Open Science]

Review History

RSOS-192068.R0 (Original submission)

Review form: Reviewer 1

Is the manuscript scientifically sound in its present form?

Yes

Are the interpretations and conclusions justified by the results?

Yes

Is the language acceptable?

Yes

Do you have any ethical concerns with this paper?

No

Have you any concerns about statistical analyses in this paper?

No

Recommendation?

Accept with minor revision (please list in comments)

Comments to the Author(s)

I see this paper for the first time, with the comments of the 4 referees from Interface attached, as well as the replies. The authors have done a very thorough job in their replies and the study is nearly acceptable as such to me, once the following further aspects are addressed:

1. You seem to equate "outliers" with "non-fitting". Outliers have a very precise definition in statistics, so please revise either your use of the term (i.e. avoid) or explain better how you use it.
2. The study is indeed made on a very large number of videos and ants, and the amount of materials in the appendices is very large too. Still, I miss some analyses. In particular, I found, hidden in one of the appendices, the fact that there is no slippage? also, I wish to have a figure, equivalent to Figure 6, where the locations of touchdowns would be plotted not according to the center of mass or the positions relative to a smooth substrate, but according to the pattern of the substrate, in particular for the 1 mm. In other words, do they adapt the exact location of the touchdown according to the very small scale features of the substrate, like within the grooves, or at the sharp edges?
3. The field study needs a bit more thinking. In which sense is it "field"? as soon as an ant enters the choice apparatus, it is not field anymore. So, I guess the field aspect is basically the uncontrolled state of the animals and temperature etc? if so, it is hardly "field" for an ecologist.
4. related to these choice experiments, I could not understand (i) how the chambers really function (i.e. where do ants enter, where is the choice made...) - This cannot be seen on figure 3, nor in S1(e) and (ii) what kind of data you really have. In particular, because you video recorded, you must know how many ants make trials in one of the arms before giving up, and hence how they walked and how many steps they made before giving up etc. The recruitment dynamics too is probably known, which might be of a different temporal scale than the one you use to clean up the substrate. Overall, it seems that this choice experiment is undervalued and underused in the paper, while it is the very best basis to discuss the more descriptive bulk of analyses, a complaint most previous referees had (and I agree)!!
5. Please have a look at the paper published since then by Humeau et al in OIB as it deals with ants walking on granular materials, a topic not equivalent but near to yours. In their discussion, they also discuss ants walking on urn plants, some of the fine scale structures being of the right scale compared to yours. Works by U. Müller, L. Gaume etc on ants walking on carnivorous plants might be indeed relevant. Related to this, I do not find the figures of ants in the wild highly relevant, as the substrate pattern bears little resemblance to the one you 3D printed.

Review form: Reviewer 2 (Christofer Clemente)

Is the manuscript scientifically sound in its present form?

Yes

Are the interpretations and conclusions justified by the results?

Yes

Is the language acceptable?

Yes

Do you have any ethical concerns with this paper?

No

Have you any concerns about statistical analyses in this paper?

No

Recommendation?

Accept with minor revision (please list in comments)

Comments to the Author(s)

Review of Uneven substrates constrain walking speed in ants through modulation of stride frequency more than stride length

This paper explores the locomotory response of ants moving over rough (uneven terrain), using an impressive sample size and automated video analysis to determine detailed changes. Overall it is an interesting paper and detailed paper which uses a broad array of techniques to answer questions which should be of broad interest to many insect biomechanists. This paper appears to have already undergone several rounds of review before I have received it. I have read through all the reviewer comments and responses, and I am satisfied with the detail of the responses and the high quality of the paper overall.

I don't have any major suggestions which could improve this paper, however I have a few minor suggestions which the authors might consider – to improve the readability of their work.

I noticed many of the reviewer comments mentioned the lack of a hypothesis driven scientific approach. The authors have acknowledged this and have modified the introduction to include a brief hypothesis, but may have sold themselves short, missing out on much of the detail that they include in the responses to reviewers.

For example

Line 77: 'We hypothesize that ant walking speed will decrease on uneven substrates correlated with similar decreases in stride frequency and stride length'

But their response on page 3 paragraph 1 was much more detailed. Why not include this in the paper? For example

We hypothesize that if ant foraging strategies aim to maximise search area/preferred speeds than walking speed should be largely invariant over different terrains. Alternatively if ant foraging strategies aim to maximise stability then we would expect walking speed to decrease, and this decrease to be affected by perturbation frequency (i.e. roughness).

Further, if a decrease in speed is apparent (as in previous studies), this decrease might just reflect a simple speed reduction strategy (where kinematics simply reflects slower speed kinematics on smooth surfaces) if a simple control strategy is used, or alternatively if a different kinematic patterns emerges it could represent a more deliberate control strategy to improve stability... (just a suggestion – not a very good one – I was just trying to include a broader testable hypothesis to your study?)

Some other minor comments

Line 105 "walking speed is slowest at the intermediate size unevenness on our substrates"

Is it really? I can see that statistically this is the case, but the distributions overlap completely, and without the effect sizes its difficult to determine how big this effect is. I think Fig 2 could be improved by giving the effect sizes in the figure caption, especially given that the authors acknowledge this problem on line 206.

Line 107 "It is unclear whether walking speed is a smooth function of ground unevenness, or if different unevenness scales induce distinct walking behaviors."

But I think you show it is not a smooth function!! This is an interesting result!

Line 118: "showing that turning is not responsible for speed reductions on uneven substrates"

Wow! What a cool result! How is that possible? What are the fore-aft accelerations vs centripetal accelerations? Most animals would slow down at high enough turn speeds. Not sure if anyone has looked at this in insects. Maybe for another manuscript though!

Inlier. I know what the authors mean, "not an outlier" - but I had never heard this term before, so I googled it and found this

Definition:

An inlier is a data value that lies in the interior of a statistical distribution and is in error. Because inliers are difficult to distinguish from good data values they are sometimes difficult to find and correct.

I don't think the authors mean their data is in error. Maybe "well-modeled data" "supported data" "standard data" "normal data" might be better?

Again, this is an interesting paper which represents a huge body of work, and I think it would be of broad interests to scientists.

Decision letter (RSOS-192068.R0)

14-Feb-2020

Dear Ms Clifton,

On behalf of the Editors, I am pleased to inform you that your Manuscript RSOS-192068 entitled "Uneven substrates constrain walking speed in ants through modulation of stride frequency more than stride length" has been accepted for publication in Royal Society Open Science subject to minor revision in accordance with the referee suggestions. Please find the referees' comments at the end of this email.

The reviewers and handling editors have recommended publication, but also suggest some minor revisions to your manuscript. Therefore, I invite you to respond to the comments and revise your manuscript.

- Ethics statement

- Data accessibility

It is a condition of publication that all supporting data are made available either as supplementary information or preferably in a suitable permanent repository. The data accessibility section should state where the article's supporting data can be accessed. This section should also include details, where possible of where to access other relevant research materials

such as statistical tools, protocols, software etc can be accessed. If the data has been deposited in an external repository this section should list the database, accession number and link to the DOI for all data from the article that has been made publicly available. Data sets that have been deposited in an external repository and have a DOI should also be appropriately cited in the manuscript and included in the reference list.

If you wish to submit your supporting data or code to Dryad (<http://datadryad.org/>), or modify your current submission to dryad, please use the following link:
<http://datadryad.org/submit?journalID=RSOS&manu=RSOS-192068>

- **Competing interests**

- **Authors' contributions**

- **Acknowledgements**

- **Funding statement**

Because the schedule for publication is very tight, it is a condition of publication that you submit the revised version of your manuscript before 23-Feb-2020. Please note that the revision deadline will expire at 00.00am on this date. If you do not think you will be able to meet this date please let me know immediately.

When submitting your revised manuscript, you will be able to respond to the comments made by

the referees and upload a file "Response to Referees" in "Section 6 - File Upload". You can use this to document any changes you make to the original manuscript. In order to expedite the processing of the revised manuscript, please be as specific as possible in your response to the referees. We strongly recommend uploading two versions of your revised manuscript:

If your manuscript is newly submitted and subsequently accepted for publication, you will be asked to pay the article processing charge, unless you request a waiver and this is approved by Royal Society Publishing. You can find out more about the charges at <https://royalsocietypublishing.org/rsos/charges>. Should you have any queries, please contact openscience@royalsociety.org.

on behalf of Dr Manoj Srinivasan (Associate Editor) and Kevin Padian (Subject Editor)
 openscience@royalsociety.org

Associate Editor Comments to Author (Dr Manoj Srinivasan):

The reviewers provide a few further minor comments to consider. One of the reviewers had suggested including at least one of the paragraphs from the previous 'response to reviewers' into the paper itself. As the reviewers suggest, we will leave it up to the authors to address these and other comments as they see fit.

Reviewer comments to Author:

Reviewer: 1

Comments to the Author(s)

I see this paper for the first time, with the comments of the 4 referees from Interface attached, as well as the replies. The authors have done a very thorough job in their replies and the study is nearly acceptable as such to me, once the following further aspects are addressed:

1. You seem to equate "outliers" with "non-fitting". Outliers have a very precise definition in statistics, so please revise either your use of the term (i.e. avoid) or explain better how you use it.
2. The study is indeed made on a very large number of videos and ants, and the amount of materials in the appendices is very large too. Still, I miss some analyses. In particular, I found, hidden in one of the appendices, the fact that there is no slippage? also, I wish to have a figure, equivalent to Figure 6, where the locations of touchdowns would be plotted not according to the center of mass or the positions relative to a smooth substrate, but according to the pattern of the substrate, in particular for the 1 mm. In other words, do they adapt the exact location of the touchdown according to the very small scale features of the substrate, like within the grooves, or at the sharp edges?
3. The field study needs a bit more thinking. In which sense is it "field"? as soon as an ant enters the choice apparatus, it is not field anymore. So, I guess the field aspect is basically the uncontrolled state of the animals and temperature etc? if so, it is hardly "field" for an ecologist.
4. related to these choice experiments, I could not understand (i) how the chambers really function (i.e. where do ants enter, where is the choice made...) - This cannot be seen on figure 3, nor in S1(e) and (ii) what kind of data you really have. In particular, because you video recorded, you must know how many ants make trials in one of the arms before giving up, and hence how they walked and how many steps they made before giving up etc. The recruitment dynamics too is probably known, which might be of a different temporal scale than the one you use to clean up the substrate. Overall, it seems that this choice experiment is undervalued and underused in the paper, while it is the very best basis to discuss the more descriptive bulk of analyses, a complaint most previous referees had (and I agree)!!
5. Please have a look at the paper published since then by Humeau et al in OIB as it deals with ants walking on granular materials, a topic not equivalent but near to yours. In their discussion, they also discuss ants walking on urn plants, some of the fine scale structures being of the right scale compared to yours. Works by U. Müller, L. Gaume etc on ants walking on carnivorous plants might be indeed relevant. Related to this, I do not find the figures of ants in the wild highly relevant, as the substrate pattern bears little resemblance to the one you 3D printed.

Reviewer: 2

Comments to the Author(s)

Review of Uneven substrates constrain walking speed in ants through modulation of stride frequency more than stride length

This paper explores the locomotory response of ants moving over rough (uneven terrain), using an impressive sample size and automated video analysis to determine detailed changes. Overall it is an interesting paper and detailed paper which uses a broad array of techniques to answer questions which should be of broad interest to many insect biomechanists. This paper appears to have already undergone several rounds of review before I have received it. I have read through all the reviewer comments and responses, and I am satisfied with the detail of the responses and the high quality of the paper overall.

I don't have any major suggestions which could improve this paper, however I have a few minor suggestions which the authors might consider – to improve the readability of their work.

I noticed many of the reviewer comments mentioned the lack of a hypothesis driven scientific approach. The authors have acknowledged this and have modified the introduction to include a brief hypothesis, but may have sold themselves short, missing out on much of the detail that they include in the responses to reviewers.

For example

Line 77: 'We hypothesize that ant walking speed will decrease on uneven substrates correlated with similar decreases in stride frequency and stride length'

But their response on page 3 paragraph 1 was much more detailed. Why not include this in the paper? For example

We hypothesize that if ant foraging strategies aim to maximise search area/preferred speeds than walking speed should be largely invariant over different terrains. Alternatively if ant foraging strategies aim to maximise stability then we would expect walking speed to decrease, and this decrease to be affected by perturbation frequency (i.e. roughness).

Further, if a decrease in speed is apparent (as in previous studies), this decrease might just reflect a simple speed reduction strategy (where kinematics simply reflects slower speed kinematics on smooth surfaces) if a simple control strategy is used, or alternatively if a different kinematic patterns emerges it could represent a more deliberate control strategy to improve stability... (just a suggestion – not a very good one – I was just trying to include a broader testable hypothesis to your study?)

Some other minor comments

Line 105 "walking speed is slowest at the intermediate size unevenness on our substrates"

Is it really? I can see that statistically this is the case, but the distributions overlap completely, and without the effect sizes its difficult to determine how big this effect is. I think Fig 2 could be improved by giving the effect sizes in the figure caption, especially given that the authors acknowledge this problem on line 206.

Line 107 "It is unclear whether walking speed is a smooth function of ground unevenness, or if different unevenness scales induce distinct walking behaviors."

But I think you show it is not a smooth function!! This is an interesting result!

Line 118: "showing that turning is not responsible for speed reductions on uneven substrates"

Wow! What a cool result! How is that possible? What are the fore-aft accelerations vs centripetal accelerations? Most animals would slow down at high enough turn speeds. Not sure if anyone has looked at this in insects. Maybe for another manuscript though!

Inlier. I know what the authors mean, "not an outlier" – but I had never heard this term before, so I googled it and found this

Definition:

An inlier is a data value that lies in the interior of a statistical distribution and is in error. Because inliers are difficult to distinguish from good data values they are sometimes difficult to find and correct.

I don't think the authors mean their data is in error. Maybe "well-modeled data" "supported data" "standard data" "normal data" might be better?

Again, this is an interesting paper which represents a huge body of work, and I think it would be of broad interests to scientists.

Author's Response to Decision Letter for (RSOS-192068.R0)

See Appendix A.

Decision letter (RSOS-192068.R1)

28-Feb-2020

Dear Ms Clifton,

It is a pleasure to accept your manuscript entitled "Uneven substrates constrain walking speed in ants through modulation of stride frequency more than stride length" in its current form for publication in Royal Society Open Science. The comments of the reviewer(s) who reviewed your manuscript are included at the foot of this letter.

You can expect to receive a proof of your article in the near future. Please contact the editorial office (openscience_proofs@royalsociety.org) and the production office (openscience@royalsociety.org) to let us know if you are likely to be away from e-mail contact -- if

you are going to be away, please nominate a co-author (if available) to manage the proofing process, and ensure they are copied into your email to the journal.

on behalf of Dr Manoj Srinivasan (Associate Editor) and Kevin Padian (Subject Editor)
openscience@royalsociety.org

Associate Editor Comments to Author (Dr Manoj Srinivasan):

Associate Editor: 1

Comments to the Author:

(There are no comments.)

Reviewer comments to Author:

Appendix A

We thank the reviewers for their many helpful comments on our manuscript. In response to their suggestions, we have added an additional supplemental figure and extended our hypotheses in the introduction. We have also changed our terminology of “inlier” strides to “well-modeled” strides. We believe that these additions improve the clarity, accuracy, and impact of our manuscript and are excited to publish in *RSOS*.

Sincerely,
Glenna Clifton
David Holway
Nicholas Gravish

Reviewer: 1

Comments to the Author(s)

I see this paper for the first time, with the comments of the 4 referees from Interface attached, as well as the replies. The authors have done a very thorough job in their replies and the study is nearly acceptable as such to me, once the following further aspects are addressed:

1. You seem to equate "outliers" with "non-fitting". Outliers have a very precise definition in statistics, so please revise either your use of the term (i.e. avoid) or explain better how you use it.

Our use of the term outliers derives from identifying points that lie >4 standard deviations from the mean in the distribution of model errors with respect to known speed values. While outliers are often identified using the interquartile range, use of standard deviations is also common. As such, we believe that our use of “outlier” is appropriate. However, thanks to reviewer feedback, we realize that the term “inlier” may be confusing and we have switched its use to “well-modeled”.

2. The study is indeed made on a very large number of videos and ants, and the amount of materials in the appendices is very large too. Still, I miss some analyses. In particular, I found, hidden in one of the appendices, the fact that there is no slippage? also, I wish to have a figure, equivalent to Figure 6, where the locations of touchdowns would be plotted not according to the center of mass or the positions relative to a smooth substrate, but according to the pattern of the substrate, in particular for the 1 mm. In other words, do they adapt the exact location of the touchdown according to the very small scale features of the substrate, like within the grooves, or at the sharp edges?

We have added this figure to the Supplemental Information (S13), however we warn about interpretations to the patterns it shows. It appears that ants prefer to step on the peaks of the checkerboard instead of the valleys, however there could be bias from the limbs being further under the body when on valleys and therefore not visible for the tracking.

3. The field study needs a bit more thinking. In which sense is it "field"? as soon as an ant enters the choice apparatus, it is not field anymore. So, I guess the field aspect is basically the uncontrolled state of the animals and temperature etc? if so, it is hardly "field" for an ecologist. Our field experiment uses wild ants under natural weather conditions across 24 hours of natural behavior without intervention from the experimenters. In this case, we believe that it is as field as many ecology experiments, including trapping and minor environmental modifications observed over time.

4. related to these choice experiments, I could not understand (i) how the chambers really function (i.e. where do ants enter, where is the choice made...) - This cannot be seen on figure 3, nor in S1(e) and (ii) what kind of data you really have. In particular, because you video recorded, you must know how many ants make trials in one of the arms before giving up, and hence how they walked and how many steps they made before giving up etc. The recruitment dynamics too is probably known, which might be of a different temporal scale than the one you use to clean up the substrate. Overall, it seems that this choice experiment is undervalued and underused in the paper, while it is the very best basis to discuss the more descriptive bulk of analyses, a complaint most previous referees had (and I agree)!!

We would love to look at the recruitment patterns and potential learning associated with the choice experiment. Unfortunately the time scale of our recording (one photo every 30 seconds) was not sufficient for tracking individual ants and recruitment. But, we are hoping to complete similar experiments in the future with higher temporal resolution to explore these questions.

5. Please have a look at the paper published since then by Humeau et al in OIB as it deals with ants walking on granular materials, a topic not equivalent but near to yours. In their discussion, they also discuss ants walking on urn plants, some of the fine scale structures being of the right scale compared to yours. Works by U. Müller, L. Gaume etc on ants walking on carnivorous plants might be indeed relevant. Related to this, I do not find the figures of ants in the wild highly relevant, as the substrate pattern bears little resemblance to the one you 3D printed.

This is an excellent paper to add! Thank you for the suggestion.

Reviewer: 2

Comments to the Author(s)

Review of Uneven substrates constrain walking speed in ants through modulation of stride frequency more than stride length

This paper explores the locomotory response of ants moving over rough (uneven terrain), using an impressive sample size and automated video analysis to determine detailed changes. Overall it is an interesting paper and detailed paper which uses a broad array of techniques to answer questions which should be of broad interest to many insect biomechanists. This paper appears to have already undergone several rounds of review before I have received it. I have read through all the reviewer comments and responses, and I am satisfied with the detail of the responses and the high quality of the paper overall.

I don't have any major suggestions which could improve this paper, however I have a few minor suggestions which the authors might consider – to improve the readability of their work.

I noticed many of the reviewer comments mentioned the lack of a hypothesis driven scientific approach. The authors have acknowledged this and have modified the introduction to include a brief hypothesis, but may have sold themselves short, missing out on much of the detail that they include in the responses to reviewers.

For example

Line 77: 'We hypothesize that ant walking speed will decrease on uneven substrates correlated with similar decreases in stride frequency and stride length'

But their response on page 3 paragraph 1 was much more detailed. Why not include this in the paper? For example

We hypothesize that if ant foraging strategies aim to maximise search area/preferred speeds than walking speed should be largely invariant over different terrains. Alternatively if ant foraging strategies aim to maximise stability then we would expect walking speed to decrease, and this decrease to be affected by perturbation frequency (i.e. roughness).

Further, if a decrease in speed is apparent (as in previous studies), this decrease might just reflect a simple speed reduction strategy (where kinematics simply reflects slower speed kinematics on smooth surfaces) if a simple control strategy is used, or alternatively if a different kinematic patterns emerges it could represent a more deliberate control strategy to improve stability... (just a suggestion – not a very good one – I was just trying to include a broader testable hypothesis to your study?)

We believe that this is a good suggestion! We have expanded our hypothesis in the introduction to expand on these topics:

Here we perform laboratory and field experiments to quantify the impact of uneven terrain on the walking kinematics and preferences of unrestrained Argentine ant workers (*Linepithema humile*). In laboratory experiments we recorded thousands of videos of ants walking on 3D-printed flat and uneven substrates with a horizontal scale approximately greater than, equal to, and less than worker body length (Fig 1a-c). In outdoor experiments we used the same substrates to test if colonies of ants would demonstrate a preference between uneven versus flat substrates. **We hypothesize that if ants use cues such as optic flow to maintain foraging productivity then net walking speed should be conserved across different terrains. Alternatively, if uneven terrain challenges walking stability, we expect ants to decrease speeds in response to the frequency of encountering perturbations and avoid terrain that induces greater instability.** To further understand how walking performance shifts on uneven terrain, we used a deep-learning approach to track limb kinematics, identifying touchdown locations and timing. We used mixed-effect modeling to identify how these gait parameters change with walking speed and compared how models trained on flat ground strides predict walking speeds on uneven substrates. **If ants use a simple control strategy for walking, we expect their limb motions on uneven terrain to kinematically resemble walking on flat ground. However, shifts in relative timing and limb positioning could represent pre-planned or feedback-induced responses to walking instability.** Our findings provide the first description of walking kinematics for ants on continuously uneven terrain while incorporating variability due to diverse, natural behaviors.

Some other minor comments

Line 105 “walking speed is slowest at the intermediate size unevenness on our substrates”

Is it really? I can see that statistically this is the case, but the distributions overlap completely, and without the effect sizes its difficult to determine how big this effect is. I think Fig 2 could be

improved by giving the effect sizes in the figure caption, especially given that the authors acknowledge this problem on line 206.

This is a good suggestion! We have added:

The percent speed reduction compared to flat ground was 42%, 41%, and 27% for 1, 3 and 5 mm substrates respectively.

Line 107 “It is unclear whether walking speed is a smooth function of ground unevenness, or if different unevenness scales induce distinct walking behaviors.”

But I think you show it is not a smooth function!! This is an interesting result!

We show that it is not a linear function, but it may still be a smooth function. We are hoping to test if there are inflection points/discrete paradigms in future work.

Line 118: “showing that turning is not responsible for speed reductions on uneven substrates”

Wow! What a cool result! How is that possible? What are the fore-aft accelerations vs centipedal accelerations? Most animals would slow down at high enough turn speeds. Not sure if anyone has looked at this in insects. Maybe for another manuscript though!

Thank you--we think it's impressive too! We do not currently have a force plate suitable for ants, but would love to test some of these questions with larger insects.

Inlier. I know what the authors mean, “not an outlier” – but I had never heard this term before, so I googled it and found this

Definition:

An inlier is a data value that lies in the interior of a statistical distribution and is in error. Because inliers are difficult to distinguish from good data values they are sometimes difficult to find and correct.

I don't think the authors mean their data is in error. Maybe “well-modeled data” “supported data” “standard data” “normal data” might be better?

Thank you for bringing this to our attention. We have changed “inlier” to “well-modeled” as you suggest.

Again, this is an interesting paper which represents a huge body of work, and I think it would be of broad interests to scientists.